# Stress-testing the resilience of the Austrian healthcare system using agent-based simulation

Michaela Kaleta [1,2,6], Jana Lasser [1,3,6], Elma Dervic[1,2], Liuhuaying Yang [1], Johannes Sorger[1], D. Ruggiero Lo Sardo[1,2], Stefan Thurner [1,2], Alexandra Kautzky-Willer [4,5] & Peter Klimek [1,2✉]

Patients do not access physicians at random but rather via naturally emerging networks of patient flows between them. As mass quarantines, absences due to sickness, or other shocks thin out these networks, the system might be pushed to a tipping point where it loses its ability to deliver care. Here, we propose a data-driven framework to quantify regional resilience to such shocks via an agent-based model. For each region and medical specialty we construct patient-sharing networks and stress-test these by removing physicians. This allows us to measure regional resilience indicators describing how many physicians can be removed before patients will not be treated anymore. Our model could therefore enable health authorities to rapidly identify bottlenecks in access to care. Here, we show that regions and medical specialties differ substantially in their resilience and that these systemic differences can be related to indicators for individual physicians by quantifying their risk and benefit to the system.

[1] Complexity Science Hub Vienna, Josefstädter Straße 39, 1080 Vienna, Austria. [2] Medical University of Vienna, Section for Science of Complex Systems, CeMSIIS, Spitalgasse 23, 1090 Vienna, Austria. [3] Graz University of Technology, Institute for Interactive Systems and Data Science, Inffeldgasse 16C, 8010 Graz, Austria. [4] Department of Internal Medicine III, Clinical Division of Endocrinology and Metabolism, Medical University of Vienna, Währinger Gürtel 18-20, A-1090 Vienna, Austria. [5] Gender Institute, A-3571 Gars am Kamp, Austria. [6] These authors contributed equally: Michaela Kaleta, Jana Lasser. ✉email: peter.klimek@meduniwien.ac.at

Practising physicians form the backbone of our healthcare systems to provide the population with access to healthcare, i.e., "the timely use of personal health services to achieve the best health outcomes"[1]. Access to healthcare might be hampered by structural barriers including the number, type, concentration and location of care providers, but also by how quickly they can be accessed[2]. Consequently, indicators that seek to quantify the access to healthcare are often derived from these metrics, such as the number of care providers per capita in a given region[3]. It was recently shown that indicators such as the density of care providers fall short of capturing structural barriers to healthcare that arise from the fact that patients do not access physicians at random[4]. Rather, physicians are embedded in informal and emergent networks of flows of patients between them[5–10]. These networks might, for instance, emerge because of geographic proximity, of one physician tending to recommend another one, or due to physicians acting as holiday substitutes to each other, and thereby encode how likely a patient will access a given care provider given that s/he has already accessed a specific other one. Such network-derived indicators for access to healthcare can paint completely different pictures on systemically important physicians with respect to quantitative indicators that solely focus on the density of care providers[4].

More concretely, using the number of physicians per capita in a region as an indicator for access to healthcare tacitly assumes that each physician is equally accessible to each individual. Given the absence of a provider, all other physicians would then be equally likely to take over additional patients. However, it has been observed that this can be a poor description of what actually happens, namely that the patients of a physician are likely to only have contact with a much more restricted set of physicians[5–8]. This gives rise to so-called patient-sharing networks that consist of care providers as nodes connected by links representing the number of patients who have had contact with both of these providers. However, it is not yet clear how to define healthcare indicators that fully acknowledge this network structure.

The functionality of the healthcare system in many developed countries is increasingly challenged by demographic shifts both in the patient population[11] and the health workforce[12] (retirement waves), climate-related risks such as extreme weather[13,14] as well as future and currently emerging infectious diseases[15]. Yet, surprisingly little attention is given to the problem of quantifying and stress-testing the resilience of national healthcare systems, as opposed to, e.g., economic or financial systems where such stress tests are a common risk management practice[16]. In this work, we aim to fill this gap by developing a stress-testing framework to quantify the resilience of primary and secondary care by physicians in Austria using agent-based simulations. As opposed to system dynamics approaches that often model flows of patients between aggregated groups of agents (e.g., different types of care providers and patients)[17,18], we adopt a data-driven individual-based approach where we track flows of patients in networks between physicians (agents). We further adopt the notion of resilience as the ability to manage and regulate the adaptive capacities of a complex adaptive system that can be described via networks[19]. In particular, we consider how regional access to primary and secondary care by physicians depends on the capacities of individual physicians when confronted with shocks that reduce the number of available physicians (be it through retirement, quarantine, or other external shocks). We model the system's response to such a shock in terms of a restructuring of the emergent patient-sharing network and quantify how likely patients are to find new care providers with sufficient capacity within a specific location and time period. The agent-based model is fully calibrated to observational healthcare access data that covers approximately 100 million visits of 7,630,498 patients at 9580 physicians of 13 different specialties in Austria.

In the following, we provide an overview of the design of the agent-based model. Details on the individual aspects of the model, such as the construction of the patient-sharing network are given in the Methods section.

Our model is comprised of two types of agents: patients that are located at a physician and physicians that have a given capacity to treat patients. Patients become displaced if their current physician is removed from the system. If a patient does not have a physician at a given point in time, they move to a new physician in the next simulation step, based on contact probabilities given in the empirically measured patient-sharing network, specified via a weighted adjacency matrix for physicians.

In addition to their capacity, physicians have one of 13 different specialties (general practitioner, urologist, etc.; see Table 1). Simulations are performed for each of the 13 specialties separately, i.e., links between physicians of different specialties have zero weight. In addition, entries in the adjacency matrix require a minimum number of shared patients $p$ to create a non-zero link between two physicians. Next to their specialty, physicians also have a location, which places them within a federal state of Austria. We perform simulations for all physicians of a given specialty and do not stratify the patient-sharing network by the federal state. We nevertheless calculate state-level indicators for all physicians and patients in a given state.

**Table 1 Descriptive table of medical specialties and their basic characteristics.**

| Specialty | Abbrev. | Physicians | Opening hours | Patients per quarter | Maximum capacity |
|---|---|---|---|---|---|
| General medicine | GP | 4967 | 19 (±6) | 2949 (±2246) | 6391 (±1825) |
| Ophthalmology | OPH | 436 | 21 (±5) | 2234 (±1955) | 4498 (±2222) |
| Surgery | SRG | 278 | 25 (±11) | 1874 (±3025) | 6495 (±3870) |
| Dermatology | DER | 290 | 21 (±4) | 2660 (±2179) | 5634 (±1722) |
| Gynaecology and obstetrics | OBGYN | 552 | 22 (±4) | 1357 (±1240) | 2887 (±1283) |
| Otorhinolaryngology | ENT | 293 | 22 (±6) | 1927 (±1235) | 4019 (±769) |
| Internal medicine | IM | 1058 | 22 (±6) | 1601 (±3424) | 4900 (±5595) |
| Paediatrics | PED | 336 | 21 (±6) | 2212 (±1788) | 4835 (±3156) |
| Neurology | NEU | 166 | 22 (±5) | 1279 (±1173) | 2681 (±1596) |
| Orthopaedics | ORTH | 366 | 22 (±7) | 4150 (±6091) | 13,522 (±5621) |
| Psychiatry | PSY | 161 | 21 (±5) | 1056 (±1115) | 2776 (±1027) |
| Radiology | RAD | 424 | 42 (±10) | 3759 (±4283) | 12,093 (±4481) |
| Urology | URO | 253 | 20 (±4) | 1558 (±1014) | 3161 (±581) |

Physicians in the data sets were divided into 1 out of 13 specialties. Numbers show mean and standard deviation. Maximum capacities $C_h$ were calculated based on opening hour information, using the top 10% of physicians per specialty.

To set the initial conditions of the simulation, each physician starts with a number of patients determined by the quarterly number of patients treated by them, specified in the empirical physician–patient contact data. In the beginning, every patient is located at a physician and no patients are searching for a new physician. After initialisation of the simulation, in each step of the simulation, all patients of all physicians that are currently not available are set to a "searching state" and an iterative displacement process commences. Within the displacement process, each patient is given $s$ opportunities (maximum number of displacement steps) to find a new physician with a free capacity to treat them. A new physician is then picked from all available physicians with a probability that is proportional to the number of shared patients between the initial and new physicians, as specified in the weighted physician adjacency matrix. In addition, patients can have a probability $\alpha$ to choose a random physician independent of the weights specified in the adjacency matrix. If the new physician is farther than a distance $d$ away from the patient's starting municipality, a new physician is picked. If after ten attempts to find a physician within distance $d$ no physician is found, the patient will go to the physician picked last, even if the physician is farther away than $d$. One displacement process constitutes one timestep of the model. Displacements are performed in random order. One timestep in the model can therefore be interpreted as the typical time between two consecutive physician visits by the same patient.

After a new physician is picked by the patient, two scenarios may occur: (1) the selected physician still has the free capacity to treat the incoming patient, the patient is no longer searching, their location is updated and the new physician's free capacity is reduced accordingly. (2) The selected physician has no free capacity to treat the incoming patient, the patient remains searching at the same location and their internal displacement counter is increased by one. The model then proceeds to the next timestep with a new re-location process. The model terminates until all searching patients were accepted by a new physician or until all searching patients have been rejected $s$ times. Patients who were rejected $s$ times are defined as lost patients and are removed from the simulation.

To test the resilience of the system, we stress it by removing random physicians. We investigate two scenarios: (i) picking a random physician of the given specialty in each artificial timestep of the simulation and removing it from the network. (ii) removing a large number of physicians in a single timestep and none thereafter. In both scenarios, physicians are selected country-wide, not limited to a certain region. Removed physicians remain unavailable for the remainder of the simulation. Scenario (i) corresponds to slow-onset shocks that play out on a timescale that is much larger than the typical time between two consecutive physician visits (e.g., retirement of physicians), whereas scenario (ii) describes fast-onset shocks that occur on much shorter timescales (e.g., unavailability due to mass quarantine in the case of an epidemic).

For scenario (i) (iterative removal of single physicians), simulation steps, i.e., removal of a single physician, are repeated until only one physician of the given specialty remains available and patients can no longer be re-located to other physicians. Based on the information of physicians and patients remaining in the system after every simulation step, we extract the state-level remaining free capacity and the number of lost patients as observables. For this scenario, an ensemble of 100 simulations is run for each parameter combination and specialty to obtain confidence intervals for the observables.

For scenario (ii) (singular large shock), only a single simulation step followed by $s$ patient displacement steps is performed. For every displacement step, we record the number of patients who currently have an active physician as the main observable and as a measure of the system's functionality. For this scenario, an ensemble of 10 simulations is run for each shock size and specialty to obtain confidence intervals for the observable.

For the first scenario, we derive two sets of indicators to measure the regional resilience of primary and secondary care sectors. One set of indicators quantifies properties related to the resilience of the entire sector (e.g., specialty) in a given region, for example, internal medicine in Vienna. The second set of indicators quantifies the properties of individual physicians. For each physician, we therefore define a risk and a benefit score. The risk score of a physician quantifies how much the removal of that physician affects other physicians' capacities (the higher this score, the more likely it is that other physicians will exceed their capacity should the given physician be removed). The benefit score, on the other hand, relates to the free capacity of a physician.

By considering the percentage of physicians with a given specialty in a region who have a risk or benefit score higher than the nationwide average, we compute regional risk and benefit levels for each specialty. Furthermore, for each region and specialty we measure the free capacity (defined as the percentage of physicians that can be removed before 20% of the initially free capacity has been filled up) and an indicator for the number of lost patients (percentage of physicians to be removed before 1% of the patients in a region do not find a new care provider in the model).

Agent-based models often produce complex and high-dimensional outputs that can be hard to interpret for both technical and non-technical experts[20]. To address this challenge, we develop a visualisation strategy for the indicators; see Fig. 1. There we show the state- and specialty-specific resilience indicators that are described by glyphs of different shapes and colours. State-specific results include aggregated values of four indicators and are depicted by a diamond-shaped glyph within a circle (a). The height and width of the diamond shape represent the states' resilience in terms of free capacity and lost patients, respectively. The colour-filled arcs within the semicircles describe the state-level percentages of physicians that have higher than state-average risk and benefit scores. The physician- or specialty-specific resilience is described by two of the previous indicators, namely risk and benefit scores (c). These are visualised by the colour-coded halves of a heart-shaped symbol that can represent the results of individual physicians as well as aggregated results of a medical specialty.

In this work, we show how to measure these indicators in a network model, see Fig. 1c, d. There we show the network neighbourhood of a physician ("physician 1") that becomes unavailable and has a relatively high-risk score, meaning that his/her removal will affect many other physicians due to an inflow of new patients (c). As a result, physician 2 with a relatively low benefit score (meaning the low capacity to accept new patients) becomes unavailable, and some patients need to contact yet another physician to find a new care provider and may potentially get lost (d).

For the second scenario, we assess the short- and long-term impact of the disruption on the system. For the former, we quantify how many patients were not able to find a new physician within one displacement step. For the latter, we count the number of patients that were not able to find a new physician after ten displacement steps and are subsequently assumed to become lost.

## Results

**Regional resilience indicators**. To investigate indicators of regional resilience of the healthcare system, we report the development of the relative lost patients as well as the remaining free capacity as physicians are gradually removed in the

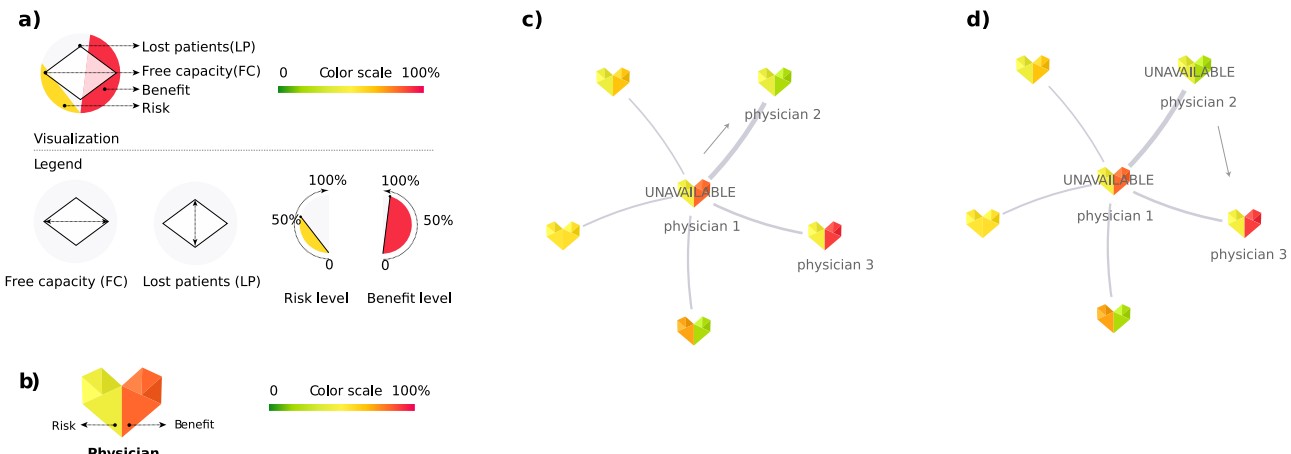

**Fig. 1 Schematic overview of visualisation elements. a** Custom glyph used to describe state-level results. **b** Custom glyph for aggregated specialty- or physician-specific resilience indicators. **c, d** Example of physicians' patient-sharing network and patient displacement: (**c**) physician 1 becomes unavailable and a share of their patients try to re-locate to physician 2. As a result, physician 2 becomes unavailable as well (**d**).

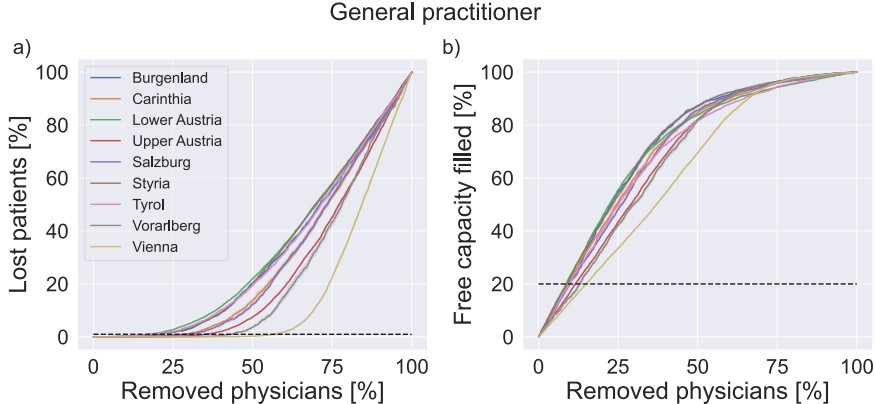

**Fig. 2 Time development of quantities of interest.** Development of lost patients (**a**) and free capacity (**b**) in federal states. Averaged model results in the example of general practitioners as physicians are gradually removed from the system. Dashed lines indicate critical limits of 1% lost patients and 20% remaining free capacity, respectively. Lines show mean values, confidence intervals are standard deviations shown as error bands (smaller than symbol size).

simulation. We report averaged results over 100 simulation iterations. Figure 2a shows (for the example of general practitioners) the development of relative lost patients and free capacity during the course of the simulation for all federal states. Initially, the removal of the first few physicians can be absorbed by the system without losing patients. At a specific percentage of removed physicians, patients start becoming "lost", e.g., unable to find another physician. The point at which the number of lost patients surpasses 1% varies widely between different states, ranging from around 20% of removed physicians (Lower Austria, Tyrol) to more than 50% (Vorarlberg, Vienna).

Figure 2 also shows how fast the initial free capacity is filled up in each federal state as GPs are gradually removed from the system. Under the sequential removal of physicians, the remaining free capacity first decreases proportionally to the number of physicians that were removed. The slopes of this linear dependence, however, vary widely across federal states and relate to the lost patients' indicator: the faster the capacity decreases, the sooner patients get lost. As the system gradually approaches a state with no remaining capacity, there is no more free capacity to accommodate displaced patients and an increasing number of patients get lost.

In Supplementary Figs. 5–16, we show lost patients and free capacity as a function of the percentage of removed physicians for

all other specialties. A summary of all states and specialties and the relative number of physicians that can be removed from the system until critical limits are reached (20% for the remaining free capacity and 1% for lost patients) is shown in Fig. 3. The resilience indicators vary widely across federal states and specialties. Resilience indicators in the state of Styria are typically on the lower side, whereas structures in Vorarlberg appear to be substantially more resilient. Psychiatrists, neurologists, surgery, internal medicine, orthopaedics and radiology show a tendency to have more resilient care networks compared to ophtalmology, dermatology, or urology. This can also be seen in Supplementary Fig. 4, where we show nationwide averages of these resilience indicators. In addition to the regional resilience of the healthcare system as a whole, we also investigate the individual risk and benefit a given physician contributes to the system. To this end, for each specialty $j$ we average the individual risk ($risk_j$) and benefit ($benefit_j$) scores over all federal states to obtain nationwide results. Supplementary Fig. 4b shows the mean and standard deviation for each specialty averaged over all federal states.

In general, there is a tendency that specialties with a high-risk score and/or a low benefit score are also less resilient in terms of lost patients and free capacity. To quantify this observation, we evaluate a linear regression model of the form

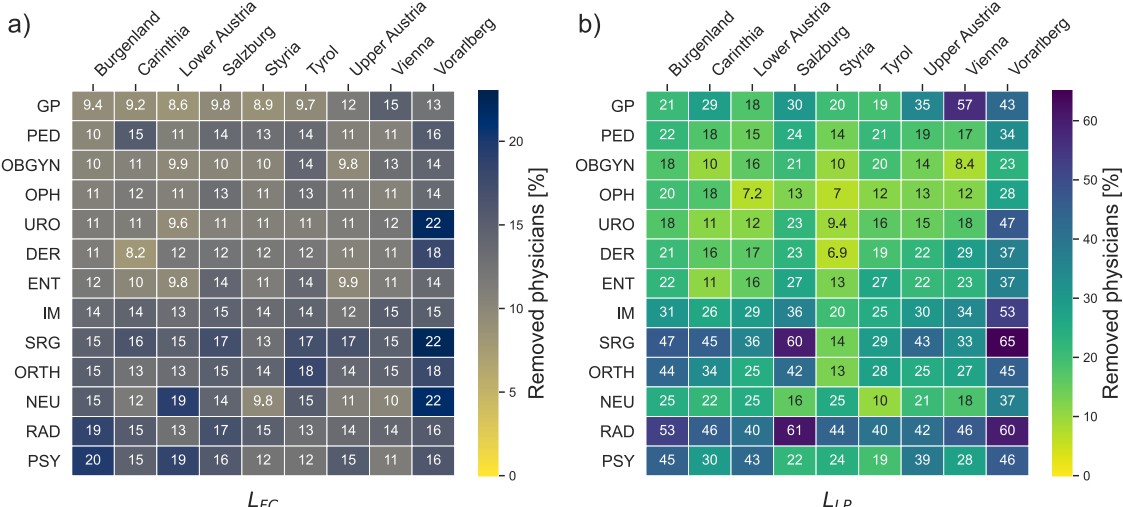

**Fig. 3 Critical resilience limits.** Average critical % of removals in federal states per specialty until limits of free capacity (**a**) and lost patients (**b**) are reached. Critical limits are 1% for lost patients and 20% for remaining free capacity. Darker colours indicate a higher number of physicians who can be removed before exceeding critical limits. Tables including the standard deviation of simulation results are given in Supplementary Tables 2 and 3. For a list of abbreviations of specialists see Table 1.

$$L_{FC_j}(L_{LP_j}) \sim r_j \cdot \text{risk}_j + b_j \cdot \text{benefit}_j + \text{const.}, \quad \text{where} \quad L_{FC_j} \text{ and}$$
$L_{LP_j}$ are the state-averaged percentages of physicians of specialty $j$ removed before reaching critical limits of free capacity and lost patients, respectively. The averaged effect sizes and standard deviations of coefficients $r$ and $b$ over all $j$ confirm the correlation between these indicators as $\langle r \rangle = -11(\pm10)$, $\langle b \rangle = 25(\pm28)$ for $L_{FC}$ and $\langle r \rangle = -70(\pm35)$, $\langle b \rangle = 58(\pm86)$ for $L_{LP}$. However, there is no discernible correlation between the risk and benefit scores across the specialties. This suggests that these scores capture two structurally independent properties that together determine the resilience of a specialty. This also motivates the use of the individual-level risk and benefit scores to assess the contributions of physicians to the systemic resilience of these regional care sectors.

**Interactive online visualisation tool**. To visualise the simulation results for the resilience indicators, we develop a visualisation strategy that can be found at https://vis.csh.ac.at/care-network-resilience/. Using the aggregated simulation data and the network structure of physicians within the healthcare system, we show detailed state- and specialty-specific information on resilience indicators, as well as the number of physicians and patients. In addition, the tool allows for the possibility of interactively removing physicians from the system whilst updating the resilience values. Exemplary results for psychiatrists in all Austrian states are shown in Fig. 4. Further detailed descriptions of the visualisation tool are shown in the SI.

**Robustness**. We performed several robustness tests to assess the sensitivity of our results. We find that varying the maximum capacity parameter $c$ ($c = 20, 30, 40$) introduced the most variation in the results. However, this variation does not change the regional resilience indicators qualitatively, meaning that the specific values for overstepping critical limits of lost patients or free capacity will differ for states when using different $c$, but it will not convert a "more resilient" state to a "less resilient" state compared to others (ranking remains robust). For the purpose of this stress test, we selected $c = 10$ as it necessitates a large proportion of physicians to increase their available capacities. Increasing the probability to approach physicians at random instead of following the links in the

patient-sharing network, i.e., setting $\alpha = 0.15$, showed the second-largest variation in simulation outcomes. Again, rankings remained robust as increasing $\alpha$ gradually washes out the network effects. In the absence of such network effects (for $\alpha = 1$) our resilience indicators would be determined by physician density alone. Other simulation parameters only show minor changes in simulation results. Every other model parameter than $c$ and $\alpha$ was tested with at least one additional setting ($s = 5, d = 50, i = 5, p = 1$) to assess robustness of the results.

**Large-scale shocks**. In addition to slow changes in the availability of physicians as described above, we also investigate a scenario in which a large portion of healthcare providers become unavailable at the same point in time. A recent survey among British physicians indicates that 10.5% of physicians were off work during the height of the SARS-CoV-2 Omicron wave in spring 2022[21]. We therefore investigate a plausible yet pessimistic scenario in which 15% of physicians become unavailable at once. We also report results for smaller (7%, 10%) and larger (20%) shocks in the SI. Figure 5 shows resilience curves after a single large-scale unavailability event for each of the 13 specialties. We note that due to the computational cost of running these simulations, we report results on ensemble sizes of 10 simulations, rather than 100 simulations, as for the previous simulations. The curves reveal interesting features of the healthcare provision networks of individual disciplines. For example, while for general practitioners, a large share of patients is able to find a new physician within a short period of time after the shock, a substantial number of patients remain without a physician and become lost after ten displacement steps. For internal medicine and orthopaedics, patients take longer to find a new physician, but almost all patients are able to find new physicians within ten displacement steps. Other disciplines such as gynaecology and obstetrics or urology have severe problems to compensate the shock, as a large number of patients are not able to find new physicians in a timely manner after the shock, and a substantial number of patients become lost after ten displacement steps. To quantify the disruption of the healthcare system due to the shock, we report the number of patients that are without a physician one and ten displacement steps after the shock in Fig. 6.

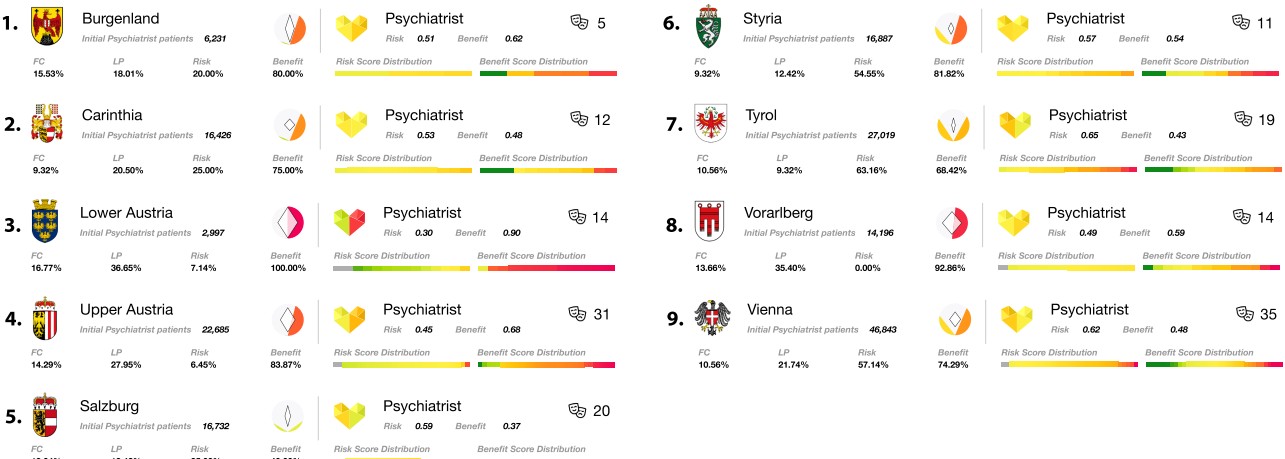

**Fig. 4 State-specific result panel.** Results for selected medical specialties (here psychiatrists) including averaged resilience indicators, numbers of patients and physicians per federal state and distributions of risk and benefit scores for each of the nine federal states.

## Discussion

Our results show that there is a limited amount of stress (number of physicians becoming unavailable simultaneously) that primary care systems and other outpatient sectors can absorb before a substantial number of patients struggle to find an alternative healthcare provider within a reasonable time. Experiences during the Omicron-variant-related surge in the SARS-CoV-2 pandemic have demonstrated that scenarios in which 10–20% of the health workforce are absent over several weeks can indeed happen[22] and need to be considered when assessing the resilience of healthcare systems. A solution could be alternative consultation methods that are less dependent on spatial proximity and could alleviate short-term shortages and help re-distribute load in the system. Especially the Covid-19 pandemic encouraged and increased the implementation of online consultations in many fields of medicine[23,24]. However, patients in regions with more deprived populations appear to be less likely to benefit from telehealth[25]. Patients with inadequate digital devices or in regions with poor internet connectivity will have problems accessing telehealth, possibly even widening already existing treatment gaps. On the other hand, remote GP but also specialist consultations may be helpful and convenient for many patients with chronic diseases in need of regular follow-up visits. However, a certain number of face-to-face consultations will remain necessary to provide adequate patients' care in the future. Therefore, although telehealth may help in allowing continued access to services in the future, we will still need a sufficient number of experienced physicians with capacities to perform online consultations in addition to standard clinical care.

Considering such a "large shock" scenario, one can think of "patients getting lost" in our model also in terms of a delayed access to health care. That is, patients might be more likely to postpone or even cancel an appointment if their usual provider is not available, or it simply might take them longer to locate a potential substitute (such as outpatient departments or other types of providers). Indeed, evidence regarding the impact of disruptions in service availability and staff absences is accumulating rapidly across multiple health outcomes even in high-income countries, including routine activities in cancer services, such as delays in radiotherapy[26], cross-sectoral services in neurological disorders[27], as well as maternal and perinatal healthcare services[28].

As healthcare demand is further expected to increase in developed countries as the population is ageing and the prevalence of chronic conditions is increasing[11], there is an increasing need to know how far the available physician supply can be stretched to still deliver the desired quality of care. In this work, we propose regional resilience indicators to quantify the point beyond which further decreases in physician densities would begin to severely impair the population's ability to access primary or secondary care. We find strong heterogeneities in these resilience indicators across different medical fields and regions. Our findings show that the naturally emerging patient-sharing networks that can be reconstructed from routinely collected administrative data are a valuable source of information when assessing the resilience of the healthcare system.

Our resilience indicators can be disaggregated into indicators for individual physicians to quantify the extent to which their hypothetical removal would stress the system (risk score) or how much of the stress from the removal of other physicians they would be able to absorb (benefit score). These indicators allow health authorities to rapidly identify physicians with sufficient capacities to take oversupply and alert them that they might receive larger than expected inflows of patients in the situation where other physicians retire. This could help planners to proactively anticipate whether such retirements could cause bottlenecks in certain regions and whether the now vacant position should be replaced with a higher level of priority. To highlight this potential of our approach, we have developed an interactive visualisation framework that allows healthcare planners and professionals to browse and intuitively access the regional and individual resilience indicators as well as to inspect these naturally emerging physician networks and how potential absences would impact them.

Our study has several limitations. Firstly, we focused on patient-sharing networks among physicians of only the same medical specialty. However, especially in rural areas, GPs are used to cover and treat a broad spectrum of diseases and they may be able to attenuate the gap caused by the shortage of some specialists[29]. On the other hand, specialists in internal medicine could temporarily replace GPs in some areas. Moreover, in Austria, many tasks of primary care are still transferred or covered by outpatient departments of bigger hospitals in the cities. Therefore, the shortfall of physicians might have different effects in rural and urban areas. Patients that get "lost" in our model might therefore manage to find a substitute in a different medical speciality. Future research might therefore explore multi-layered stress-test models, in which each layer corresponds to one specialty and where we also consider patient flows across different layers. In addition, primary care

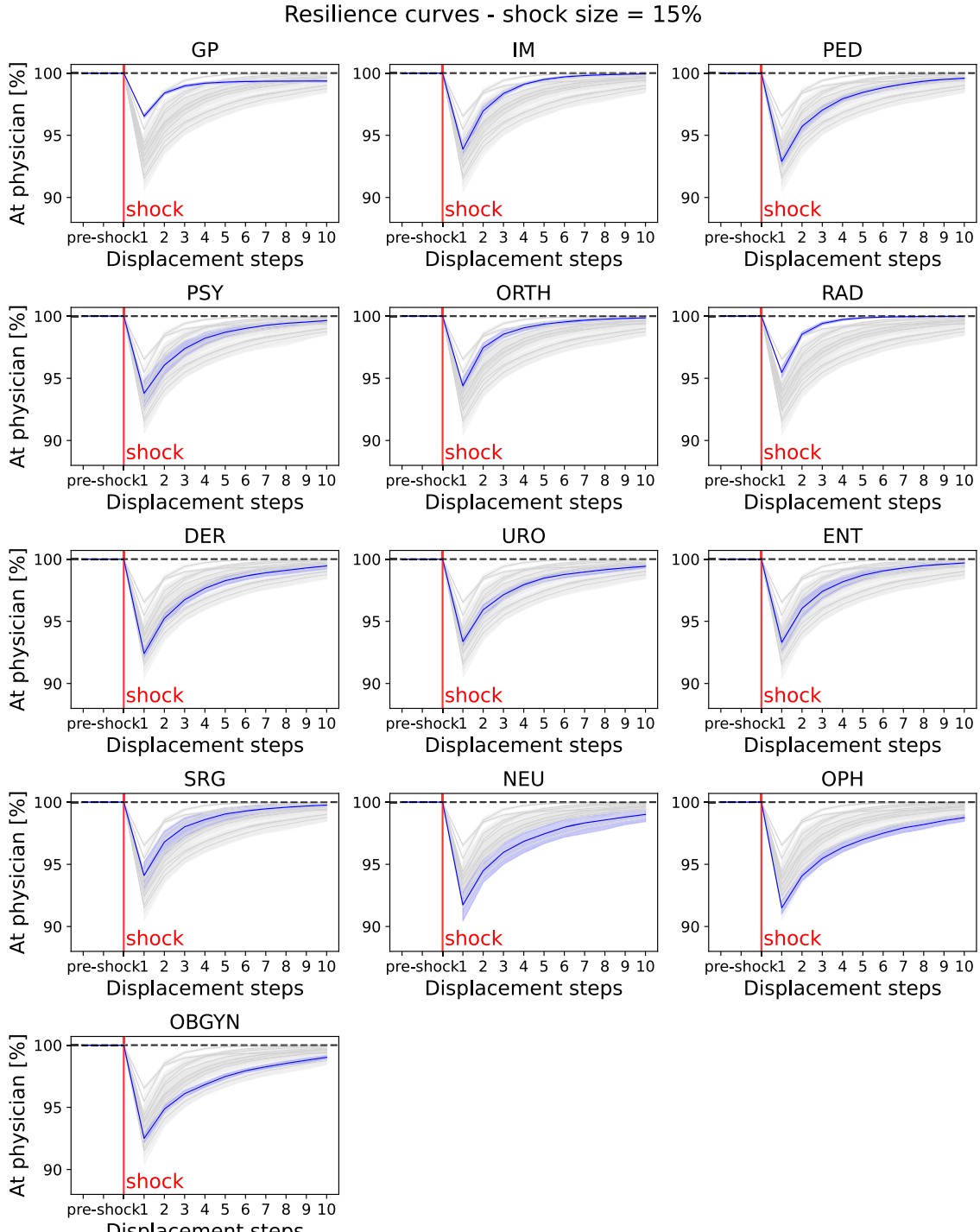

**Fig. 5 Large-scale unavailability event.** Individual panels show resilience curves for each specialty (for a list of abbreviations of specialists see Table 1). The curves show the number of patients that have a physician over ten displacement steps after a single large shock where 15% of randomly chosen physicians of the given specialty were removed simultaneously. Single plots highlight one specialty while others are shown in grey for comparison. Lines and error bands display mean and standard deviation over an ensemble of ten simulations. The dashed black line represents no patients on active lookout for a new physician.

centres that host a range of different medical specialists and which are currently being established in Austria might change the resilience characteristics. On the one hand, physicians might be able to easier substitute for their colleagues. On the other hand, such centres might increase the risk of multiple physicians failing at once, for example in the case of an infectious disease that spreads within the centre.

Secondly, the results of the agent-based simulation and particularly the concrete value of, e.g., the risk and benefit scores, depend on the assumptions regarding various model parameters. We performed several robustness tests to better understand the sensitivity of our results. We find that our results are most sensitive to changes in the maximum capacity parameter $c$ and probability of random re-locations $\alpha$. However, we observed no

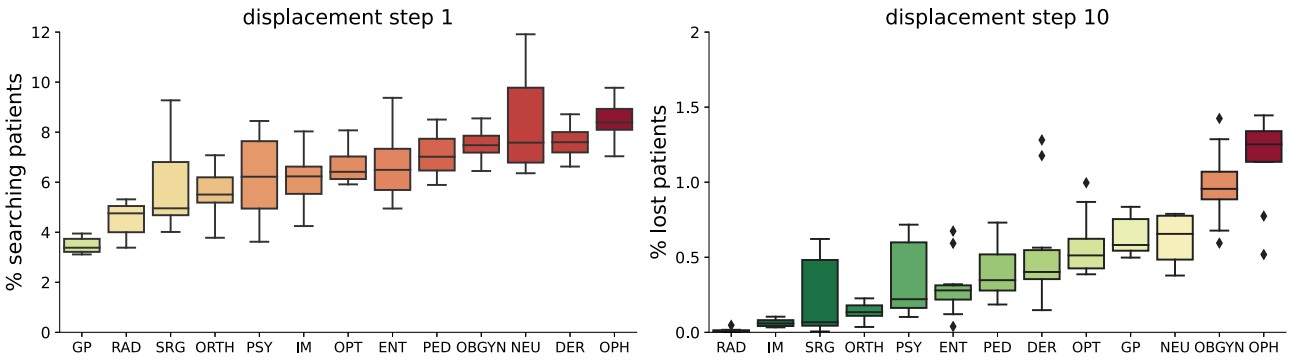

**Fig. 6 Impact of a large-scale unavailability event on searching and lost patients.** Percentage of patients looking for a new physician for each specialty one and ten displacement steps after a large shock where 15% of physicians of a given specialty were removed from the network. Patients who remain searching after 10 timesteps become lost. Boxes show the quartiles of the data set of an ensemble of 10 simulation runs, whiskers show the rest of the distribution. Points over 1.5 of the inter-quartile range past the high and low quartiles are considered outliers and shown as individual points. For a list of abbreviations of specialists see Table 1.

substantial changes in the rankings of states and specialties, respectively.

Thirdly, the model should include more complex behaviours of doctors in future analyses. Recent trends show that the number of physicians with care obligations is steadily increasing, leading to different requirements regarding work-life balance, working models, fields of activities and duties and the need for family-friendly workplaces[30].

Fourth, even in universal care systems access to physicians may also depend on socio-economic characteristics of the patients, whereas our approach only captures heterogeneity amongst the physicians. Further improvements to the stress test methodology need to be considered to understand whether socially disadvantaged groups are more likely to "get lost" and which interventions might alleviate this issue.

Our work demonstrates that the resilience of primary and specialist care is an emergent property of formal and informal networks that physicians form amongst themselves. While fiscal constraints and concerns regarding an ageing physician workforce in rural areas are growing worldwide[31], we find that these care networks indeed possess tipping points in terms of their ability to provide care to the entire population. Removing or losing an alleged excess capacity or oversupply of physicians might inadvertently push healthcare systems closer to their tipping points. Using our indicators allows health authorities to quantify how close they are to these tipping points and thereby strike a more data-informed balance between system resilience and effectiveness.

## Methods
The presented research complies with all ethical regulations. Since the present study re-uses existing administrative data, the requirement for an assessment by an ethics review board was waived, after consideration of the study design by the Commission for Scientific Integrity and Ethics of the Graz University of Technology.

The Federal Law on Documentation in the Health Care System in Austria provides the legal basis for written informed consent not being required for this study: It allows the documentation of health-related data in the intra- and extra-mural outpatient and inpatient care sectors, as well as for the processing of patients' and service providers' data in pseudonymized form for certain purposes including (long-term) monitoring of epidemiological developments relevant to health policy as well as the implementation and further development of integrated health structure planning and health services research.

**Data**. To construct the patient-sharing network, we combine two data sets: patient contacts for Austrian physicians as derived from administrative data, and a data set of physician's opening hours that was scraped from a healthcare information platform (www.herold.at) in March 2020 (permission for scraping the data was obtained from the owners of the platform). Some physicians practice more than one medical specialty and/or practice in more than one municipality. Therefore, in

the following, we always refer to a unique (physician, specialty, municipality) combination, when talking about the number of physicians. A physician who is both, say, a general practitioner and an internist, or a physician who operates as a general practitioner in two municipalities is counted as two different physicians. We note that the latter—a physician that practices in two locations—can lead to a situation where a physician is removed in one location but keeps practising in the second. This is unrealistic in a situation where the physician ceases to practise because of illness. This simplification does not introduce a large error to our model, since only a small number of physicians (4.3%) practice in more than one location and would be affected by this artefact.

**Patient contacts**. The health record data provided by the Austrian Federal Ministry of Health contains all recorded patient visits (approx. 103,000,000) over a 1-year period from 1 January 2018 to 31 December 2018. The data contain patients' contacts with physicians from 13 different medical specialties and includes a pseudonymised unique ID for every patient and physician, the date of contact as well as the locations of physicians in form of 5-digit municipality IDs. The first digit of the municipality ID is further used to determine the federal state (9 states in total). The data set contains 9580 unique combinations of (physician, specialisation, and municipality).

**Patient-sharing network**. To model connections between physicians, we use the patient contact data to construct a patient-sharing network. In this network, every node is a physician. The network is described by the adjacency matrix $A_{i,j}$, where entries $a_{i,j}$ describe the number of shared patients between physicians $i$ and $j$, i.e., the number of patients that visited both physician $i$ and physician $j$ in a given period of time. The adjacency matrix is therefore symmetrical by definition and the network described by the adjacency matrix is undirected. To construct the adjacency matrix, we start with a patient's visit to physician $i$ and consider all visits of the given patient to other physicians during a period of 3 months before and after the initial visit to physician $i$. All thus identified physicians are therefore sharing one patient with $i$. This process is repeated for all patient visits of physician $i$. If there is at least one shared patient between $i$ and a given other physician $j$, there is an edge $a_{i,j}$ between $i$ and $j$. The edge is weighted by the sum of shared patients. This procedure is repeated for all physicians, resulting in a weighted undirected patient-sharing network that is represented by the adjacency matrix $A_{i,j}$. To reduce spurious relations, the adjacency matrix can be thresholded by requiring a minimum number of shared patients and a maximum geographical distance between physicians $i$ and $j$. The distance is measured as the direct distance between $i$'s and $j$'s municipalities. The location of municipalities was determined as either the geolocation of the affiliated municipal office or as the geolocation of the area with the highest population density in the municipality using a population density map. If the distance between physicians $i$ and $j$ exceeds threshold $d$, the edge is removed and $a_{i,j} = a_{j,i} = 0$. Similarly, when the number of shared patients between $i$ and $j$ is smaller than threshold $p$, the edge is removed. To model the patient-sharing network within a medical specialty, only the subset of nodes corresponding to physicians of the given specialty and their corresponding edges are used. Edges between physicians of different specialties are removed.

**Capacity estimation**. To estimate a physician's maximum capacity, $C$, we aggregate the patient contact data with information about opening hours; see SI Section "Matching of patient contact data to opening hours" for details. We provide estimates of the capacity based on the opening hours of physicians and patient contacts. For this opening-hour-based capacity estimate $C_i$, we divide physicians of every specialty into bins based on their total number of weekly opening hours with a bin size of 5 opening hours. For each bin, we then identify the top 10% (for

baseline setting) of physicians in terms of their capacity (number of quarterly patient contacts) and calculate their median capacity. This median capacity is then assigned to all remaining physicians in the bin as their maximum patient capacity.

By merging the information of the two data sets, we get a comprehensive description of the majority of physicians in the country. We can initially describe each physician with specific characteristics: the specialty, municipality, number of patients seen per quarter ($N$) and total capacity based on opening hours ($C_h$). While the specialty and location are constant, the current number of patients and the remaining free capacity change during simulations. General descriptive information is shown in Supplementary Table 1. The vast majority of physicians are general practitioners ($n = 4967$), while psychiatrists ($n = 161$) form the smallest specialty group. Patient variables in the model include their starting location (municipality) based on the physician they start at, the number of displacement steps they have already undergone, step-wise travelled distance and total travelled distance.

**Model parameters**. The model contains a number of parameters that can be varied and tested for robustness. We define a baseline parameter setting as follows: The maximum number of displacement steps before a patient is considered lost ($s = 10$), the median patient capacity of top c% of the most visited physicians of a certain specialty that is assigned to all other physicians as their maximum capacity ($c = 10$), the maximum distance in kilometres patients are willing to travel to a new physician ($d = 100$), the minimum number of shared patients in the adjacency matrix for a valid connection ($p = 2$), and the probability $\alpha$ of random re-locations of patients to unconnected physicians (which in the baseline setting does not occur, $\alpha = 0$). Physicians who are not connected to the giant component[32] of the patient-sharing network remain in the system for the possibility of patients to choose them at random (only if $\alpha > 0$).

**Free capacity and lost patients**. In the scenario where physicians are iteratively removed from the system, the remaining free capacity in each federal state is one of the main observables in our model. The initial free capacity in each state is reduced in each simulation step as more and more physicians become unavailable. We track how the remaining relative free capacity in each state is filled up/diminished until there is no capacity left and all the physicians are removed from the system. We define an arbitrary critical limit of 20% of the remaining free capacity per state and track the average relative number of physicians that needs to be removed to reach this critical free capacity limit, $L_{FC}$. Similarly, we sum the number of lost patients and calculate the average relative number of cumulatively lost patients in each federal state. The more physicians are unavailable, the lower the remaining free capacity and the higher the probability for a patient to get rejected by a new physician—after all physicians are removed, 100% of patients are lost. Analogous to the free capacity, we define an arbitrary critical lost patient limit of 1% and track the relative number of physicians removed before this point is reached, $L_{LP}$. For the scenario in which a single large shock is applied to the system, the number of lost patients is recorded after a single simulation step, which includes ten patient displacement steps as patients are searching for a new physician.

**Risk and benefit scores**. For the scenario in which physicians are iteratively removed from the system, we define a risk and benefit score for each physician, based on the negative and positive influence that the physician contributes to the healthcare system. The risk score $R_i$ represents physician $i$'s risk and is measured by the average extra load of patients that their first-degree neighbours in the patient-sharing network must bear in case of $i$'s unavailability. We therefore define the risk score as $R_i = \left\langle \min\left(\frac{N_j + N_i \cdot w_j}{C_j}, 1\right) \right\rangle_j$, where $N_i$ and $N_j$ stand for the number of patients of physician $i$ and their neighbours $j$. $C_j$ is $j$'s total capacity and $w_j$ describes the normalised connection weight between $j$ and $i$. The range of the risk score is $[0, 1]$, where 0 corresponds to the lowest and 1 to the highest risk. The benefit score $B_i$ represents physician $i$'s beneficial contribution to the healthcare system in terms of their initial free capacity. The benefit sores are normalised over all physicians of a given medical specialty and range as well $[0, 1]$, with 0 being the lowest benefit and 1 the highest benefit.

**Reporting summary**. Further information on research design is available in the Nature Research Reporting Summary linked to this article.

## Data availability

The simulation data generated in this study have been deposited in OSF under accession code https://doi.org/10.17605/OSF.IO/H5E9A. The raw and processed patient contact data are not available due to privacy laws. The data set is safeguarded by the Austrian Federal Ministry of Health and made accessible to research institutions under strict data protection regulations. To gain access to these data, researchers have to find individual arrangements with the Austrian Federal Ministry of Health. We publish a sample data set under accession code https://doi.org/10.5281/zenodo.6576023 to showcase how the patient-sharing network is created from such data and how simulations are run. The opening hour data are publicly available at herold.at. The data collected in March 2020

for this study are available upon reasonable request from the corresponding author. Data can be shared for research purposes. Requests will be considered for 10 years after the publication of this article.Data used in this study can be provided by the Austrian National Public Health Institute: Gesundheit Österreich GmbH, Stubenring 6, 1010 Vienna, kontakt@goeg.at. After reviewing the request, data will usually be provided within 4 weeks.

## Code availability

Python 3.8.5 was used to perform the simulations and data analysis. The simulation and analysis code for this study is available under MIT license in the repository at https://doi.org/10.5281/zenodo.6576023.

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

## Acknowledgements

M.K., A.K.-W. and P.K. acknowledge financial support from the Medizinisch Wissenschaftlicher Fonds des Buergermeisters der Bundeshauptstadt Wien under CoVid004. J.S. and S.T. acknowledge financial support from the Austrian Science Promotion Agency FFG under 882184. J.L. acknowledges financial support by the Marie Skłodowska-Curie grant no. 101026507.

## Author contributions

M.K., J.L., S.T. and P.K. designed research; M.K., J.L. and D.R.L.S. performed research; J.S. and L.Y. contributed analytic tools; M.K., J.L., E.D., D.R.L.S., J.S. and L.Y. analysed data. All authors reviewed and contributed to the manuscript.

## Funding

## Competing interests

The authors declare no competing interests.
