## [Peer Review File · Nature Communications]

Reviewers' Comments:

Reviewer #1:

Remarks to the Author:

The authors present a stress-test of the healthcare system in Austria. The problem tackled is of clear importance. The paper is well written and the narrative clear. The platform developed to explore the results is very interesting, intuitive and easy to navigate. A clear plus.

The authors decided to limit quite a bit the details about the framework proposed in the main. I think the real novelty of the paper is the stress-test model which, as the authors correctly point out, is used in finance, but not yet in other critical systems such as healthcare. Unfortunately, the details are left out the main and only summarized in the methods.

The application of the framework to the case of Austria is interesting and the results point, as one could have somehow imagined, to heterogeneities in risks and resilience. However, it is rather unclear how realistic the "stress-test" actually is, and how the results presented are bounded by the context where they are obtained (i.e., Austria). When I started reading the paper, I was expecting a focus on the incredible disruption that COVID created, but the results come from data from 2018 and again the stress test is based on (random?) removal of doctors. Just to draw a comparison, papers based on the resilience of power grids often use famous real outages as applicative examples, also suggesting possible ways to improve it. Here, the "exercise" is disconnected from any previous shocks to the system. I still see the value of what is done, but clearly the results would be way more interesting and a definitely fit to Nat Comm in case real scenarios would be discussed.

The authors speak about "structural barriers" to healthcare access, there are many some of which connected to socio-economic status that are still visible even in universal care systems.

The importance of "network thinking" in this context is clear to people with experience in networks, but considering a broader readership this point could be made more explicit (final part of the first paragraph of the intro)

It is unclear how doctors are removed. At random? By region? The details of the "stress" should be described. Also, in many definitions of resilience (none of which is actually used in the paper) time is a crucial factor. In the paper, similarly to what is done in classic network damage papers, doctors (nodes) are removed sequentially. However, the resilience is also dependent of the time the system has to adapt/react. Losing 10 doctors in 1 week is different than losing 10 in 1 months. The way time is considered here is not clear.

What happens in reality to "lost" patients? Do they go to ER?

The risk and benefit of each doctor is computed in the unperturbed network as function of the effect each doctor has on the workload of others. This assumes however that patients are assigned to the other doctors connected to the one removed. What would happen in real cases? I guess this depends on the country, but in Austria what would happen? It is unclear and the way shocks are dealt with is the key aspect of any stress-test.

The choice of 20% and 1% as threshold is unclear.

Figure 1 condenses many of the results of the paper. It is very dense and presented very early, before many of the details are discussed. I would suggest splitting the figure and add one before to describe the framework and the test-test.

Why do we have 100% benefit for lower Austria if the risk semicircle is full and the benefit semicircle empty?

The scale of the distribution of risks and benefits for profession are not really easy to interpret. Benefit is the free capacity of physician, so why we have another quantify in the glyphs called FC (free capacity)?

What are those two faces (the icon) in the speciality plot?

When comparing different regions would be interesting to see in the x a quantity that allows comparisons. I imagine the region of the capital is the most densely populated, so maybe instead the number of doctors you could consider the % respect to the total in each region? Also, how the dependencies across regions are considered? Can patients look for doctors in other regions? in many countries patients travel quite a bit for healthcare, is this considered here? These points are discussed briefly in the methods where we learn that there is a cutoff of 100km that each patients is happy to travel before getting "lost". Is this a realistic number? In many countries people choose or have to travel more than that.

In line 117 the word removed is repeated

In figure 2, in some cases we see confidence intervals, in others these are very small. Why such a small variation in places like vienna? The order of removal or the type of doctor removed is not really important?

In figure 3, y labels are not really helpful to understand the specialties

In figure 3, the choice of color highlights more resilient places rather than those more at risk. I would imagine showing the less resilient places is more important

Figure 4 is very hard to read and interpret

Is there any reason to expect a linear relationship between the two metrics and risk/benefit?

The initial part of the discussion is very relevant but is rather disconnected from what is presented before.

The authors do not try to offer any possible solution to deal with shocks, optimal ways to re-route patients thus improving the resilience of the system.

Are the networks of different specialties disconnected? So, there are not referrals?

The authors used one year of data from 2018. Few points here. Is this enough data to get a good picture? Is 4 year data still a good description of the system especially after covid?

Reviewer #2:

Remarks to the Author:

Recommendations and comments overall:

Introduction

The goal of this paper is to present a model for understanding the effect on access to healthcare when physician availability decreases. This presents a way for health systems to explore how patient flows are affected by HRH fluctuations and is noteworthy given the experiences many health systems have faced during the COVID-19 pandemic. It also is important for many rural areas that may very well share health resources.

The application of agent-based simulation to stress-test the Austrian health system is based on constructs available in other sectors such as finance/economics and on previous work by others in the field. The authors provide a useful demonstration of the application of this approach and provide evidence that this tool can be added to the arsenal of health workforce planning tools. While not within the scope of this work, looking at how this simulation modeling approach dovetails with system dynamic modeling – one is focused on agent availability and networks, the other focused on patient flows.

Results

Overall, the language is necessarily complex as statistical modeling is not generally conducive to simplification and will be clear to others in this specific field. While much of the language needs to

be mathematical the authors do attempt to put examples and descriptions in simpler terms to allow a more lay person to understand the nature and usefulness of the model.

The visualization strategy is a critical component to communicating the results of the simulation and kudos to the authors for thinking this through. The visualization presented in the figures were helpful to see how results would be communicated to decision-makers, although not immediately user-friendly. This may be more of a result of having to read it, rather than experiencing it during a larger communication (i.e., reading a visualization is not as good as having it presented visually during a discussion/presentation of results).

Discussion

While the examples presented in the paper only applies the stress test to physician networks (physician to physician links), can this model be used to explore other models of care – i.e., if a physician is not available, can patients seek alternate professionals such as Nurse Practitioners. A discussion of applications of this simulation model using broader networks may contribute more to the generalizability of the model beyond physician-centric networks.

Methods

Data: When describing the calculations of number of physicians, the authors indicate that if a physician has roles in different capacities or regions, then they would be counted as two different physicians in the modelling. How does this impact the simulation – i.e., if they are physician 1 in one locale but physician 4 in another, does the model remove # 4 in the second locale when they are removed in the first locale (given that they are not available to patients); or does the model assume that while this physician becomes unavailable in the first locale, they remain available in the second locale, which would not make sense if the physician is removed due to illness?

What are the noteworthy results?

The results from this work highlight the importance of understanding access, capacity and continuity of care especially in the context of COVID and virtual care.

Will the work be of significance to the field and related fields? How does it compare to the established literature? If the work is not original, please provide relevant references.

Access to primary care is a high priority area and the methodology and findings from this work can inform policy and decision making in a very important way in Canada and globally although findings are within context of Austria. The limitations acknowledged are important and will need careful consideration and additional work to be meaningfully applied within practice settings. For instance, capacity estimation as acknowledged by authors, GPs in rural settings do operate in a different way and provide comprehensive primary care, and patients on the panel have to be considered within the context of complexity of care needs.

Authors also acknowledge the importance of the COVID learnings and virtual care context. Another consideration is how collaborative team settings can be considered and the contribution of other professionals in adding capacity.

Does the work support the conclusions and claims, or is additional evidence needed?

This is a great starting work and as authors have acknowledged will need additional work to follow this important contribution for findings to be meaningfully applied within practice context. How patient complexity, virtual care, rural comprehensive primary care, collaborative team members add capacity for access.

Are there any flaws in the data analysis, interpretation and conclusions? Do these prohibit publication or require revision?

Data analysis methods are sufficient within the context of this work. Limitations are well acknowledged by authors and need for additional work. This can be published. I would recommend considering other limitations that have been suggested.

Is the methodology sound? Does the work meet the expected standards in your field?

The methodology is sound and expected standards in the field have been met.

Is there enough detail provided in the methods for the work to be reproduced?

Yes there is enough detail provided in the methods for work to be reproduced. Limitations that have been acknowledged are critical for this work to be meaningfully reproduced.

Reviewer #3:

Remarks to the Author:

The authors attempted to address an interesting problem. They used ABM for testing the resilience of the Austrian healthcare system. In doing so, they used rigorous datasets from two sources: administrative and physician opening hours. However, the manuscript has some serious flaws, and they need to be addressed to improve the merit of this work. Please see below for my more specific comments.

1. Consideration of patient-sharing networks only captures only one side of the scenario. The authors should consider the other side of the scenario – physician sharing patient networks. A patient's degree in such a network will demonstrate visits to a common physician. This consideration is necessary to improve the generability of the findings.
2. ABM has its own limitation (e.g., not suitable for homogeneous data). From different illustrations, it seems that data are not heterogeneous. Again, the authors need to analyse the scenario using the network disintegration and robustness approaches to cross-validate their findings further.
3. The concept of a 'patient-sharing network' demands a proper reference in the manuscript
4. Some of the figures in the main manuscript carry little information (e.g., Figure 1). There are too many figures in the supplementary materials. Again, they do not carry much information.
5. Not sure whether the authors considered the demands for specialist physicians (e.g., cancer, palliative care, etc.). Are their reduction approach (take off one physician from the network) random? Do you consider the necessity of specialist physicians in special settings? If not, their findings would not do any good for people (e.g., cancer patients) who need the healthcare service most.

Response to reviewers: Stress-testing the Resilience of the Austrian Healthcare System Using Agent-Based Simulation

We thank all three reviewers for their valuable feedback, which helped us improve the content and clarity of our manuscript. In response to the reviewers's feedback, we introduced two main changes to the manuscript: (1) we added a second "shock" scenario, in which a large share of physicians is removed at once, similar to the current situation in the ongoing pandemic where a large number of physicians becomes unavailable due to illness or quarantine. (2) we significantly expanded the description of our modelling approach in the introduction, in the hope of improving the overall clarity of our manuscript. We provide point-by-point responses to individual reviewer's comments below. Original reviewer's comments are indicated in *italic* font.

Reviewer #1

The authors present a stress-test of the healthcare system in Austria. The problem tackled is of clear importance. The paper is well written and the narrative clear. The platform developed to explore the results is very interesting, intuitive and easy to navigate. A clear plus.

(1) The authors decided to limit quite a bit the details about the framework proposed in the main. I think the real novelty of the paper is the stress-test model which, as the authors correctly point out, is used in finance, but not yet in other critical systems such as healthcare. Unfortunately, the details are left out the main and only summarized in the methods.

We have taken the reviewer's advice to heart and have substantially expanded our description of the model in the introduction section (lines 73-135).

(2) The application of the framework to the case of Austria is interesting and the results point, as one could have somehow imagined, to heterogeneities in risks and resilience. However, it is rather unclear how realistic the "stress-test" actually is, and how the results presented are bounded by the context where they are obtained (i.e., Austria). When I started reading the paper, I was expecting a focus on the incredible disruption that COVID created, but the results come from data from 2018 and again the stress test is based on (random?) removal of doctors. Just to draw a comparison, papers based on the resilience of power grids often use famous real outages as applicative examples, also suggesting possible ways to improve it. Here, the "exercise" is disconnected from any previous shocks to the system. I still see the value of what is done, but clearly the results would be way more interesting and a definitely fit to Nat Comm in case real scenarios would be discussed.

We appreciate the feedback by the reviewer and think that assessing the effect of large-scale shocks on the system is a valuable addition to our work. We therefore conducted additional simulations where we removed a large share (7%, 10%, 15% and 20%) of physicians of a given specialty at once and investigated the response of the system. The shock sizes are informed by reports of unavailability of physicians in Great Britain during the height of the latest Omicron wave¹. As a result of this additional work, we now differentiate between two types of strain on the system: (i) slow removal of individual physicians, resembling for example retirement and (ii) a large shock to the system in form of the simultaneous removal of a significant share of the physicians, resembling for example isolation and quarantine of infected or exposed physicians during an epidemic. We have introduced this differentiation in the introduction (lines 115-123) and added a new section to the results, where we report our findings (lines 238-255 and new figures 5 and 6). We also clarified in the introduction that physicians are removed at random (line 115).

(3) The authors speak about “structural barriers” to healthcare access, there are many some of which connected to socio-economic status that are still visible even in universal care systems.

We agree that our framework does not yet acknowledge that socially disadvantaged groups might be more likely to “get lost” than other groups. We have included this now in the discussion (lines 330-334) as an open question, to not only add heterogeneity to the physician population but also to take into account different socio-economic strata of the population.

(4) The importance of “network thinking” in this context is clear to people with experience in networks, but considering a broader readership this point could be made more explicit (final part of the first paragraph of the intro)

We have added a paragraph in the introduction (lines 43-51) where we seek to make this network thinking more explicit to the “in-terms-of-networks-uninitiated reader”.

(5) It is unclear how doctors are removed. At random? By region?

We have clarified this point in our newly added general model description in the introduction (lines 88-93).

(6) The details of the “stress” should be described. Also, in many definitions of resilience (none of which is actually used in the paper) time is a crucial factor. In the paper, similarly to what is done in classic network damage papers, doctors (nodes) are removed sequentially. However, the resilience is also dependent of the time the system has to adapt/react. Loosing 10 doctors in 1 week is different than loosing 10 in 1 months. The way time is considered here is not clear.

One timestep in the model can be defined as the typical timespan (population average) between two consecutive contacts with a physician of the same specialty. Shocks can

¹ <https://www.rcplondon.ac.uk/news/rcp-survey-finds-one-ten-doctors-work>

therefore be classified according to whether they occur on a timescale that is faster or slower than two consecutive visits. For a “fast” shock (think mass quarantine in an epidemic), multiple removals occur simultaneously in the model, whereas for “slow” shocks (e.g., population aging) removals take place sequentially and the system responds adaptively to each individual removal. To reflect this, we have now introduced a “fast” and a “slow” time scale for the stress applied to the system by introducing a second type of stress that we apply to the system. We now differentiate between the “slow” stress that is applied to the system by removing one physician in each simulation step (as described in the previous version of this manuscript), and the “fast” stress that is applied to the system by removing a substantial number of physicians simultaneously (as also described in our reply to point (2)). We now introduce and describe these two types of stress in the introduction (lines 115-123) and give analogies to “real” scenarios such as physicians leaving the workforce (“slow” shock), and physicians becoming infected or having to quarantine because of an epidemic (“fast” shock). We also introduce a new measure of system functionality, namely the number of patients that are at an active physician, to quantify the effects that fast stress has on the system (lines 168-171).

(7) What happens in reality to “lost” patients? Do they go to ER?

We have extended the discussion of how “lost patients” can be thought of particularly in the context of a COVID scenario along these lines, together with a discussion of recent literature on service disruptions (lines 276-283). There is indeed ample literature on this. One of the challenges in the context of the COVID shock is that particularly early in the pandemic some patients might have canceled visits not due to service unavailability but due to fear of visiting a doctor’s office. We have, therefore, constrained the literature to works that explicitly acknowledge service unavailability or staff absence as the cause for the disruption. Furthermore, we already discussed the limitations that some patients that get lost in the model might indeed end up in different specialties, but did not make this connection explicit. We have added a statement to further clarify this (lines 308-312).

(8) The risk and benefit of each doctor is computed in the unperturbed network as function of the effect each doctor has on the workload of others. This assumes however that patients are assigned to the other doctors connected to the one removed. What would happen in real cases? I guess this depends on the country, but in Austria what would happen? It is unclear and the way shocks are dealt with is the key aspect of any stress-test.

We implemented an additional analysis based on the available original dataset to detect physicians who stopped working during the year and therefore act as examples of real-life physician removals. Patients of physicians who were active in the first quarter (Q1) of the year and became inactive afterwards were tracked in the fourth quarter of the year (Q4). For each specialty we ascribe a connection between the inactive physician in Q1 and the next visited physician of same speciality in Q4. We compare the resulting real-life connections to the links in the model’s adjacency matrix to assess the portion of patients using a non-existing, weak or strong link. We observe that in these real-life removal scenarios most patients in fact utilized links with high weights (>97.5%), as shown in the figure below.

(9) *The choice of 20% and 1% as threshold is unclear.*

These thresholds are indeed chosen arbitrarily. We clarified this in the text at the appropriate place.

(10) *Figure 1*

(a) *Condenses many of the results of the paper. It is very dense and presented very early, before many of the details are discussed. I would suggest splitting the figure and add one before to describe the framework and the test-test.*

Following the reviewer's advice we have split the figure into two separate figures. The first figure now includes panels (a) to (d) from the original figure and introduces the visualization strategy for the resilience indicators. The second figure (figure 4 in the manuscript) includes panel (e) from the original manuscript and describes state-level resilience results.

(b) *Why do we have 100% benefit for lower austria if the risk semicircle is full and the benefit semicircle empty?*

We thank the reviewer for spotting this issue. The semi-circles were mis-labeled, we corrected this in the new version of the manuscript.

(c) *The scale of the distribution of risks and benefits for profession are not really easy to interpret.*

We agree. One reason for this is that the exact value of these risk and benefit scores depends on model parameters that can be hard to validate against real world evidence. However, what remains robust across a broad range of model specifications are the rankings in these scores. Hence, our strategy of visualizing and communicating these results focus more on these rankings as discussed as the second limitation. We have clarified there that the results for the risk and benefit values (as opposed to the ranks) are subject to this limitation (lines 324-325).

(d) Benefit is the free capacity of physician, so why we have another quantify in the glyphs called FC (free capacity)?

We distinguish between free capacity as a state-level indicator and a physician-level indicator. We hope that the reworked Figure 1 makes this distinction clearer.

(e) What are those two faces (the icon) in the speciality plot?

It is a symbol representing the specialty of psychology. The number next to it shows the number of specialist physicians in that federal state.

(11) When comparing different regions would be interesting to see in the x a quantity that allows comparisons. I imagine the region of the capital is the most densely populated, so maybe instead the number of doctors you could consider the % respect to the total in each region?

We note that we already report numbers with respect to the total in a given state.

(12) Also, how the dependencies across regions are considered? Can patients look for doctors in other regions? in many countries patients travel quite a bit for healthcare, is this considered here? These points are discussed briefly in the methods where we learn that there is a cutoff of 100km that each patients is happy to travel before getting "lost". Is this a realistic number? In many countries people choose or have to travel more than that.

We consider the full network of physicians of a given specialty in all of Austria and do not separate regions in our simulations. Therefore, patients are free to cross state borders to look for a new physician. We clarify at the appropriate place in the introduction (lines 86-89).

We measured the specialty-specific average travelled distance of patients throughout the simulations (see exemplary figures for GPs and surgeons below) and see that based on the connections in the adjacency matrix, patients on average do not need to travel further than the specified maximum of 100km. On the contrary, most patients travel much less than that (see above examples of GP and surgery). With increased numbers of unavailable doctors, patients need to travel further until a point where the number of lost patients increases rapidly (see "noise" in regions above 70% removed physicians).

Travelled distance GP.

Travelled distance surgeons.

(13) In line 117 the word removed is repeated

We thank the reviewer for spotting this error and have corrected it in the revised manuscript.

(14) In figure 2, in some cases we see confidence intervals, in others these are very small. Why such a small variation in places like vienna? The order of removal or the type of doctor removed is not really important?

Confidence intervals are smaller than the width of the line - we added this statement in the figure caption. This indeed means that the order of doctor removal has limited impact on the outcome for some combinations of specialties and regions.

(15) In figure 3, y labels are not really helpful to understand the specialties

We added a reference to Table 1 in the figure caption where the abbreviations of specialties are defined.

(16) In figure 3, the choice of color highlights more resilient places rather than those more at risk. I would imagine showing the less resilient places is more important

We have reversed the colormaps on both heat-maps to highlight states and specialties that are less resilient compared to others.

(17) Figure 4 is very hard to read and interpret

We agree with the reviewer. Since the same results are already reported in the second part of what used to be figure 1 (now figure 4 in the manuscript), we removed this figure as it was redundant.

(18) Is there any reason to expect a linear relationship between the two metrics and risk/benefit?

Indeed, as for the case of the calculation of risk and benefit scores the number of patients is conserved, we would expect a linear relationship between the two metrics.

(19) The initial part of the discussion is very relevant but is rather disconnected from what is presented before.

We agree with the reviewer on this point. Given our newly added results for a second “large shock” scenario (see also our response to point (2)), we now connect this point of the discussion to the challenges the healthcare system experienced during the Covid-19 pandemic.

(20) The authors do not try to offer any possible solution to deal with shocks, optimal ways to re-route patients thus improving the resilience of the system.

We agree that identifying optimal ways of re-routing patients or points at which physicians could be added to increase the resilience of the system would be a very valuable contribution to the discussion. We nevertheless think that the present work is valuable in that it lays the foundation for such investigations in the future. As such, we think that research into optimal solutions is out of scope of the current work but are eager to continue our research in this direction in the future.

(21) Are the networks of different specialities disconnected? So, there are not referrals?

Yes, networks of different specialties are disconnected - we now clarify this at the appropriate place in the introduction (lines 82-84). We think that this is the appropriate way to deal with the patient sharing network as our present work aims to investigate substitutions, not referrals.

(22) The authors used one year of data from 2018. Few points here. Is this enough data to get a good picture? Is 4 year data still a good description of the system especially after covid?

We agree that it will be highly interesting to follow-up on this work with data that covers the time period of the pandemic. Nevertheless, as discussed in the manuscript, we strongly believe that the framework applies to a host of relevant scenarios, one of them being pandemic shocks. It is also relevant to establish a pre-pandemic baseline. There is also the practical bottleneck that an updated physician dataset (particularly covering the recent Omicron surge, where this issue of high staff absences was most visible during the pandemic so far) is currently not available. As the editor also noted that it is not necessary to bring new data for this revision, we believe it to be beyond the scope of this manuscript to enlarge the data basis. Also note that in the absence of shocks there will be no rapid restructuring in these networks. The overall number of physicians changed from 46337 in 2018 in Austria to 47224 in 2019 and 47674 in 2020, i.e. percentage-wise in low single digits per year. Processes driving this change are mostly migration of the workforce, retirement and the entry of new physicians. Of course, we strongly agree that these changes on a slow time scale might add to substantial burdens over longer periods of times (as discussed), we

believe that using 2018 data is sufficient to establish a baseline for analysing the COVID shock a few years later.

Reviewer #2

The goal of this paper is to present a model for understanding the effect on access to healthcare when physician availability decreases. This presents a way for health systems to explore how patient flows are affected by HRH fluctuations and is noteworthy given the experiences many health systems have faced during the COVID-19 pandemic. It also is important for many rural areas that may very well share health resources. The application of agent-based simulation to stress-test the Austrian health system is based on constructs available in other sectors such as finance/economics and on previous work by others in the field. The authors provide a useful demonstration of the application of this approach and provide evidence that this tool can be added to the arsenal of health workforce planning tools.

- (1) While not within the scope of this work, looking at how this simulation modeling approach dovetails with system dynamic modeling – one is focused on agent availability and networks, the other focused on patient flows.*

We have added a statement in the introduction (lines 59-62) where we briefly delineate approaches using agent-based models (i.e., models in which one element corresponds to one agent) from system dynamics approaches (as models that study dynamics on a macro level typically between different classes of agents).

- (2) The visualization strategy is a critical component to communicating the results of the simulation and kudos to the authors for thinking this through. The visualization presented in the figures were helpful to see how results would be communicated to decision-makers, although not immediately user-friendly. This may be more of a result of having to read it, rather than experiencing it during a larger communication (i.e., reading a visualization is not as good as having it presented visually during a discussion/presentation of results).*

We thank the reviewer for this feedback on our visualization strategy. Also in response to other reviewer's comments (see reviewer #1, point (10)) we have improved the clarity of our visualizations, namely by splitting Figure 1 into two parts: the first part, which remains to be figure 1, illustrates our general visualization strategy. The second part, which is now figure 4 in the results section, describes our results in the framework of our visualization strategy.

- (3) While the examples presented in the paper only applies the stress test to physician networks (physician to physician links), can this model be used to explore other models of care – i.e., if a physician is not available, can patients seek alternate professionals such as Nurse Practitioners. A discussion of applications of this simulation model using broader networks may contribute more to the generalizability of the model beyond physician-centric networks.*

We have extended the discussion section along these lines (lines 308-312). In particular we discuss that “lost patients” might actually end up with alternate professionals. This could be modelled using a multilayer network approach, in which each layer corresponds to a specialty and patients might also “flow” from one layer to another. The development of such models might not only be interesting to extend the stress-test framework, but of course also to address questions related to multidisciplinary coordination across different care sectors.

(4) Data: When describing the calculations of number of physicians, the authors indicate that if a physician has roles in different capacities or regions, then they would be counted as two different physicians in the modelling. How does this impact the simulation – i.e., if they are physician 1 in one locale but physician 4 in another, does the model remove # 4 in the second locale when they are removed in the first locale (given that they are not available to patients); or does the model assume that while this physician becomes unavailable in the first locale, they remain available in the second locale, which would not make sense if the physician is removed due to illness?

We run simulations separately for each specialty. Therefore, a physician that has two specialties, they will appear both in the simulation for the first specialty, as well as in the simulation for the second specialty. Since simulations for different specialties do not interact with each other, removing the physician in one of the simulations has no impact on the other simulation. We clarified this point at the relevant place (lines 82-88).

In a case where the same physical practices are in two locales, we would indeed only remove them from one locale while keeping them at the second. We agree that this is not realistic in the case where a physician is removed due to illness. We recognize this is a limitation of our model but nevertheless think that this simplification is still warranted, as only a small number of physicians (4.3%) practices in two locations. We now give a statement to this effect at the appropriate place in the methods section (lines 352-356)

(5) The results from this work highlight the importance of understanding access, capacity and continuity of care especially in the context of COVID and virtual care. Access to primary care is a high priority area and the methodology and findings from this work can inform policy and decision making in a very important way in Canada and globally although findings are within context of Austria. The limitations acknowledged are important and will need careful consideration and additional work to be meaningfully applied within practice settings. For instance, capacity estimation as acknowledged by authors, GPs in rural settings do operate in a different way and provide comprehensive primary care, and patients on the panel have to be considered within the context of complexity of care needs. Authors also acknowledge the importance of the COVID learnings and virtual care context. Another consideration is how collaborative team settings can be considered and the contribution of other professionals in adding capacity.

We agree that the effect of team settings would be interesting to consider in further work, as it might both positively (easier substitution) and negatively (easier spread of an infectious disease between physicians) affect the system’s resilience. We add a statement to that effect at the appropriate place in the discussion (lines 314-319).

This is a great starting work and as authors have acknowledged will need additional work to follow this important contribution for findings to be meaningfully applied within practice context. How patient complexity, virtual care, rural comprehensive primary care, collaborative team members add capacity for access.

Are there any flaws in the data analysis, interpretation and conclusions? Do these prohibit publication or require revision?

Data analysis methods are sufficient within the context of this work. Limitations are well acknowledged by authors and need for additional work. This can be published. I would recommend considering other limitations that have been suggested.

Is the methodology sound? Does the work meet the expected standards in your field?

The methodology is sound and expected standards in the field have been met.

Is there enough detail provided in the methods for the work to be reproduced?

Yes there is enough detail provided in the methods for work to be reproduced. Limitations that have been acknowledged are critical for this work to be meaningfully reproduced.

Reviewer #3

The authors attempted to address an interesting problem. They used ABM for testing the resilience of the Austrian healthcare system. In doing so, they used rigorous datasets from two sources: administrative and physician opening hours. However, the manuscript has some serious flaws, and they need to be addressed to improve the merit of this work. Please see below for my more specific comments.

(1) Consideration of patient-sharing networks only captures only one side of the scenario. The authors should consider the other side of the scenario – physician sharing patient networks. A patient's degree in such a network will demonstrate visits to a common physician. This consideration is necessary to improve the generability of the findings.

We thank the reviewer for the valuable consideration but do not agree that this is relevant for the application testing the functionality of the healthcare network. The analogy to failing nodes in the patient sharing networks (physicians becoming unavailable) would be patients becoming unavailable. We do not think that this would impact the functionality of the healthcare system, which is the focus of this work. We agree that a physician sharing network would be an interesting perspective to take but think it is out of scope for the present work.

(2) ABM has its own limitation (e.g., not suitable for homogeneous data). From different illustrations, it seems that data are not heterogeneous. Again, the authors need to analyse the scenario using the network disintegration and robustness approaches to cross-validate their findings further.

Physicians are highly heterogeneous in their attributes, namely their capacity, number of patients they serve and spatial distribution. We therefore do not agree that our data are not heterogeneous and remain convinced that an agent-based model is a suitable approach to the problem at hand. We recognise that ABMs have limitations but do not see them as crucial for our application. The ABM has been developed such that the vast majority of agent (physician) properties can be directly observed in data. The remaining (few) parameters that cannot be directly measured have been subjected to comprehensive robustness tests as described in the corresponding “Robustness” section in Results and value constructive feedback on how these robustness tests could be further clarified.

(3) The concept of a ‘patient-sharing network’ demands a proper reference in the manuscript

We have added a paragraph in the introduction (lines 43-51) where we explicitly introduce the concept of a patient-sharing network.

(4) Some of the figures in the main manuscript carry little information (e.g., Figure 1).

We split Figure 1 into an explanatory part discussed in the introduction, and a state-specific results panel part for the results section. In addition, we removed what used to be Figure 4 in the original manuscript, since the information was partly redundant with Figure 1. We chose to keep panels (a)-(d) of figure one, since our visualization strategy is an integral part of making our research results actionable for practitioners. We note that we were specifically complimented for our visualization strategy by other reviewers.

(5) Not sure whether the authors considered the demands for specialist physicians (e.g., cancer, palliative care, etc.). Are their reduction approach (take off one physician from the network) random? Do you consider the necessity of specialist physicians in special settings? If not, their findings would not do any good for people (e.g., cancer patients) who need the healthcare service most.

We clarify our approach at randomly removing physicians from the system at the appropriate place in the introduction (lines 115-123). We consider each of the 13 specialties separately and run simulations for different settings for all of them. To not overload the main manuscript, we provide the corresponding results in the SI (figures S5 to S16).

(6) There are too many figures in the supplementary materials. Again, they do not carry much information.

Most of the material reported in the supplement pertains to either robustness checks (as requested in comment 2 by the reviewer), which we think are crucial to ensure the validity of our simulation approach, or results for individual specialties. As pointed out by the reviewer in point 5, results for individual specialties such as internal medicine or urology might be of interest for a smaller audience while not providing additional insights over the key results of our modelling work provided in the main manuscript. We therefore choose to provide these results in an online-only supplement for interested readers.

Reviewers' Comments:

Reviewer #1:

Remarks to the Author:

The authors did a great job revising the manuscript. They have addressed all my concerns and/or provided sound responses.

I have only two minor points to mention.

It is not clear why the authors run only 10 simulations for the "fast shock" while they run 100 in the original scenario.

In figure 5 the authors show averages and single runs results. In some cases the averages looks very far from the "mass" of the single runs. Are the plots correct?

Reviewer #2:

Remarks to the Author:

The authors' thoughtful and fulsome response to the reviews are appreciated. They have addressed my comments adequately. I would note that on lines 98 and 221, there are grammatical errors which need to be fixed during the editing phase.

Reviewer #3:

Remarks to the Author:

The revised manuscript does not address all my concerns, especially about the network construction and heterogeneity. There is a difference between network-level heterogeneity and attribute-level heterogeneity. The manuscript does not demonstrate a good understanding of network construction and how to cross validate the underlying findings from multiple network construction angles.

Reviewer response for “Stress-testing the Resilience of the Austrian Healthcare System Using Agent-Based Simulation”

Reviewer #1

It is not clear why the authors run only 10 simulations for the “fast shock” while they run 100 in the original scenario.

We report results on ensemble sizes of 10 simulations, rather than 100 simulations due to the computational cost of running these simulations. We added a statement to this end at the appropriate point in the results section.

In figure 5 the authors show averages and single runs results. In some cases the averages looks very far from the “mass” of the single runs. Are the plots correct?

We think this is a misunderstanding on the side of the reviewer: the individual panels of figure 5 show result for a single specialty (line highlighted in blue) over the results for all other specialties (grey lines). Therefore, the results for a single specialty are not to be expected to present the average of the grey lines.

Reviewer #2

The authors' thoughtful and fulsome response to the reviews are appreciated. They have addressed my comments adequately. I would note that on lines 98 and 221, there are grammatical errors which need to be fixed during the editing phase.

We thank the reviewer for spotting these and have corrected the grammatical errors.

Reviewer #3

The revised manuscript does not address all my concerns, especially about the network construction and heterogeneity. There is a difference between network-level heterogeneity and attribute-level heterogeneity. The manuscript does not demonstrate a good understanding of network construction and how to cross validate the underlying findings from multiple network construction angles.

As discussed with the editor, we would need more specific details regarding the reviewer's critique to be able to address specific portions of our work.